# Safety of hyperbaric oxygen therapy in non-emergent patients with a history of seizures: A retrospective cohort study

Subin Park[1], Anton Marinov[2,3], Hance Clarke[2,3,4], Simone Schiavo[2,4], Elise Greer[2], George Djaiani[2,4], Jordan Tarshis[5], Rita Katznelson[2,3,4]*

1 Temerty Faculty of Medicine, Institute of Medical Science, University of Toronto, Toronto, Ontario, Canada, 2 Hyperbaric Medicine Unit, Toronto General Hospital, Toronto, Ontario, Canada, 3 Rouge Valley Hyperbaric Medical Center, Scarborough, Ontario, Canada, 4 Department of Anesthesiology & Pain Medicine, University of Toronto, Toronto, Ontario, Canada, 5 Department of Anesthesia, Sunnybrook Health Science Centre, Toronto, Ontario, Canada

* rita.katznelson@uhn.ca

**Data Availability Statement:** All relevant data are within the manuscript.

**Funding:** The author(s) received no specific funding for this work.

## Abstract

### Background

Hyperbaric oxygen therapy (HBOT) is well established as a treatment for various medical conditions. However, it poses a risk of oxygen toxicity, which can cause seizures particularly in individuals with pre-existing seizure disorders. Consequently, seizure disorders are considered a relative contraindication to HBOT. Despite this, the relative risk of HBOT-induced seizures in this patient population remains unclear. This retrospective cohort study aims to evaluate the safety of HBOT among patients with pre-existing seizure disorders.

### Methods

After obtaining approval from the Research Ethics Board, we retrospectively reviewed the patient charts of individuals with a history of seizures who were referred to the Rouge Valley Hyperbaric Medical Center and Toronto General Hyperbaric Medicine Unit for HBOT between January 2020 and December 2023. Relevant demographic information, past medical history, and HBOT session treatment protocols, such as the treatment pressure set in absolute atmospheric pressure (ATA) and number of air breaks, were recorded. The collected data was analyzed using descriptive statistics.

### Results

A total of 43 patients were referred to HBOT during the study period, and 21 patients did not proceed with the treatments. In total, 634 HBOT sessions were administered to 22 patients in monoplace chambers with five-minute air breaks, and one patient experienced a seizure event. Each patient completed an average of 29 (range 3–60) sessions lasting 90–120 minutes at 1.8 ATA (n = 3), 2.0 ATA (n = 18), or 2.4 ATA (n = 1). Fifteen patients were on oral antiseizure medications during the HBOT course. The overall incidence of seizures was one in 634 treatments.

**Competing interests:** Rita Katznelson, Anton Marinov and Hance Clarke are shareholders of the Rouge Valley Hyperbaric Medical Center. This does not alter our adherence to PLOS ONE policies on sharing data and materials.

## Conclusion

While patients with a history of seizures may develop seizure activity during HBOT, the majority can safely undergo treatment when predetermined protocols are followed. With careful management and adherence to established protocols, HBOT can be a viable treatment option for those with seizure histories.

## Introduction

Hyperbaric Oxygen Therapy (HBOT) involves breathing 100% oxygen under pressure greater than normal atmospheric pressure. It can be conducted in monoplace (or single patient) chambers or in a multiplace chamber, with several individuals together. HBOT enhances the healing of inflammatory and microcirculatory disorders by increasing oxygen availability, antioxidant activity, vasoconstriction, and angiogenesis [1]. Generally considered a safe therapy, HBOT is commonly indicated for a variety of conditions, including diabetic foot ulcers, chronic non-healing wounds, radiation injuries, and necrotizing infections, among others [2]. However, the therapy is not without potential side effects, which can include ear barotrauma, claustrophobia, and, in rare cases, seizures [3].

Seizures during HBOT are a known complication with a relatively low incidence—as low as 0.25 in 10,000 to 6 in 10,000 treatments [4–13]. These seizures are often attributed to oxygen toxicity, where high concentrations of oxygen inhaled at elevated pressures lead to disturbances in the central nervous system [14, 15]. Importantly, even patients without a prior history of seizures can develop oxygen-induced seizures [16]. However, other potential causes that can mimic oxygen-induced seizures, such as hypoglycemia, side effects of medications that lower the seizure threshold, and pre-existing brain pathology, should also be considered [17].

The pathophysiology of oxygen toxicity seizures is not fully understood. One potential mechanism involves elevated levels of oxygen- and nitrogen-derived free radicals with the increase in partial pressure of oxygen during HBOT, which can cause cell membrane lipid peroxidation, enzyme inhibition, and alteration of nucleic acids and protein synthesis [15, 18, 19]. Additionally, increased nitric oxide (NO) production, stimulated by HBOT, may contribute to excess excitatory neurotransmitter (glutamate) activity [20–22]. These antioxidant responses, combined with an imbalance in vaso-modulation, are believed to influence oxygen toxicity seizures [21]. Another possible mechanism is the effect of hyperbaric oxygen on brain glucose transport and metabolism. Hyperoxia decreases AMP (adenosine monophosphate) concentrations in endothelial cells of the blood brain barrier (BBB) and in cells behind the BBB. This suppression reduces the activity of glucose transporters responsible for moving glucose from the blood into cerebrospinal fluid (CSF) and from the CSF to brain cells. The cumulative effect of the diminished glucose concentration in the CSF and reduced glucose transport to the cells beyond the BBB results in local hypoglycemia, hypoglycemic brain syndrome, and seizures [23].

Oxygen induced seizures generally manifest as generalized tonic-clonic convulsions, although focal seizures have also been reported. These seizures may occur suddenly or be preceded by an aura [24, 25]. The risk of seizures is influenced by many factors such as increased treatment pressure, and conditions that could decrease the seizure threshold, such as carbon dioxide retention, brain tumour or brain soft tissue radionecrosis, hypoglycemia, hyperthyroidism, carbon monoxide poisoning, and certain medications [4, 7–9, 16, 26]. While some

consider history of seizures as a risk factor for oxygen-induced seizures [16, 27], others suggest no incremental increase in risk [9]. Referrals of patients with a history of seizures to hyperbaric units are not uncommon, especially due to the comorbidity between seizure disorders and conditions typically managed with HBOT, such as diabetic foot ulcers [28]. Excluding these individuals from treatment without robust evidence of heighted risk may inadvertently lead to suboptimal care. Thus, understanding HBOT safety profile in patients with seizure history is essential for guiding clinical decision-making and obtaining informed patient consent for HBOT.

This retrospective cohort study aims to evaluate the application of HBOT in patients with seizure histories who have undergone HBOT for non-emergent indications. The study seeks to provide insights into its safety, which may inform future research and contribute to refining treatment protocols.

## Methods

### Study design

This retrospective cohort study included patients with a history of seizures who underwent non-emergent HBOT between January 2020 and December 2023 at one of the two following hyperbaric centres in Ontario, Canada: Rouge Valley Hyperbaric Medical Centre, Scarborough and Toronto General Hyperbaric Medicine Unit, Toronto. Institutional Research Ethics Board approvals (CAPCR ID: 24–5426 and Veritas IRB Inc 2024-3523-17786-2.) were obtained to collect data from patients' medical records (last access to data on August 9, 2024).

### Participants and data collection

Patients 18 years of age or older with a documented history of seizures who underwent non-emergent HBOT during the study period were included. A history of seizures was defined as one or more provoked or non-provoked seizure events obtained from medical records. Additionally, the participating hyperbaric centres approved HBOT for patients who had been seizure-free for at least three months, and those with poorly controlled seizure disorders were determined on a case-by-case basis but generally not approved to initiate therapy. Information collected from patients' charts included age, sex, past medical history, medications, date of last seizure, indication for HBOT, duration of treatment sessions, prescribed treatment pressure, number of air breaks, and details of any seizure events during the course of the therapy. All patients provided written consent to receive HBOT for a clinical indication approved by Health Canada.

### Hyperbaric oxygen therapy protocol

Patients were placed in a monoplace chamber (3600H/HR or 4100H, Sechrist Industries, Inc. Anaheim, CA, USA; or Sigma 36, Perry Baromedical Hyperbaric Medical Systems, Riviera Beach, Fl, USA) compressed on 100% oxygen at 1.8, 2.0, or 2.4 absolute atmospheres (ATA) for 90–120 minutes, with 1–3 five-minute air breaks per session. During the air breaks, air (21% oxygen) was supplied through a non-rebreather face mask attached to a hospital air supply that is controlled by a flow meter on the outside of the chamber.

### Outcomes

The main objective of this study was to describe the safety profile of consecutive HBOT sessions among non-emergent patients with a prior history of seizures. The main outcome included the incidence of seizures during HBOT in this patient population.

### Statistical analysis

Descriptive statics was used to summarize qualitative data, such as patient demographics and history of seizures. Continuous data were reported as means ± standard deviations.

## Results

### Clinical data

Of the 43 patients referred to HBOT, 21 patients did not receive HBOT due to refusal to participate in HBOT (n = 11); poorly controlled seizure disorder, defined as an episode of seizures in the last three months despite anti-seizure therapy (n = 3); palliative state (n = 3); severe confinement anxiety (n = 1); multiorgan dysfunction (n = 1); inability to commute to the treatment centre (n = 1); or referral to another center (n = 1). Twenty-two patients with a documented seizure history (Table 1) were included in the final analysis.

The mean patient age was 58 ± 15 years, and nine (41%) were female (Table 1). The most common comorbid conditions included hypertension (8; 36%), history of substance use (5; 23%), and chronic obstructive pulmonary disease/asthma/interstitial lung disease (4; 18%). Fifteen (68%) patients were on antiseizure medications, five (23%) were on opioid therapy, and four (18%) were on steroid medications.

Most patients had a documented etiology for the seizures: five patients (23%) had seizures related to structural abnormalities such as intracranial tumour and brain radiation necrosis; four patients (18%) had substance withdrawal/abuse-related seizures; and three patients (14%) had seizures related to infection and metabolic derangements. The remaining 10 patients (45%) had a seizure history of unknown etiology (Table 2). Indications for HBOT were most commonly for complex wounds (6; 27%), idiopathic sensorineural hearing loss (4; 18%), and osteomyelitis (3; 14%). Other indications for HBOT included diabetic foot ulcers (2; 9%), osteoradionecrosis (2; 9%), radiation cystitis (2; 9%), and soft tissue radiation injury (radiation-induced brain necrosis) (2; 9%). Two patients with brain radiation necrosis (#20, 22, Table 2) had seizures three months prior to HBOT including the patient (#22, Table 2) with ongoing focal aware seizures. Both patients underwent HBOT uneventfully.

**Table 1. Patient baseline demographics, co-morbidities, and medications.**

|  | n = 22 |
| --- | --- |
| Age, years (average ± SD) | 58 ± 15 |
| Female | 9 |
| **Comorbidities** |  |
| Radiation-induced brain necrosis | 3 |
| COPD/asthma/ interstitial lung disease | 4 |
| Coronary artery disease | 2 |
| Diabetes mellitus | 3 |
| History of substance use | 5 |
| Hypertension | 8 |
| Liver cirrhosis | 2 |
| Thyroid disease | 2 |
| **Medications** |  |
| Antiseizures | 15 |
| Opioids | 5 |
| Steroids | 4 |

Abbreviations: COPD, chronic obstructive pulmonary disease; SD, standard deviation.

**Table 2. History of seizures and HBOT protocol.**

| Patient ID # | Indication for HBOT | Seizure history | Last seizure episode (years prior to HBOT) | Current antiseizure therapy | ATA/ number of air breaks | Total number of treatments |
|---|---|---|---|---|---|---|
| 1 | Complex wound | Single seizure, ethanol withdrawal | 3 | No | 2.0/1 | 13 |
| 2 | Complex wounds | Seizures of unknown etiology x2 | 27 | Yes | 2.0/2 | 37 |
| 3 | ISSHL | Shock-induced seizures | 23 | No | 2.0/1 | 30 |
| 4 | Complex wound | Single seizure, ethanol withdrawal | 1 | Yes | 2.0/1 | 50 |
| 5 | DFU/OM | Cocaine-induced seizure/acute subdural hematoma | 10 | No | 2.0/2 | 3 (refused to continue) |
| 6 | OM | Seizure of unknown etiology | 10 | Yes | 2.0/2 | 43 |
| 7 | DFU | Sepsis-induced seizure | 4 | No | 2.0/2 | 4 (refused to continue) |
| 8 | OM | Benign brain tumor-induced seizure | NA | Yes | 2.0/2 | 40 |
| 9 | ISSHL | Seizure of unknown etiology | 6 | Yes | 2.0/2 | 20 |
| 10 | ORN | Focal seizures | 3 | Yes | 1.8/2 | 60 |
| 11 | Radiation cystitis | Seizure of unknown etiology | 2 | Yes | 2.0/2 | 5 (refused to continue) |
| 12 | ISSHL | Seizure of unknown etiology | 20 | No | 1.8/2 | 20 |
| 13 | Radiation cystitis | Hyponatremia-induced seizure | 3 | No | 2.0/1 | 40 |
| 14 | Complex wound | Hepatic encephalopathy/ ethanol withdrawal | 0.5 | Yes | 2.0/2 | 10 |
| 15 | ORN | Seizure of unknown etiology | 6 | Yes | 2.0/2 | 40 |
| 16 | Soft tissue radiation injury | Brain radiation necrosis-induced seizure | 0.5 | Yes | 2.0/3 | 38 |
| 17 | OM | Seizure of unknown etiology | 2 | Yes | 2.4/2 | 20 |
| 18 | ISSHL | Seizure of unknown etiology | 12 | No | 2.0/2 | 20 |
| 19 | Complex wound | Meningioma-induced seizures | 1 | Yes | 2.0/2 | 40 |
| 20 | OM | Brain radiation necrosis-induced seizure | 0.25 | Yes | 2.0/3 | 18 |
| 21 | Complex wound | Seizure of unknown etiology | 1 | Yes | 2.0/2 | 53 |
| 22 | Brain radiation necrosis | Brain radiation necrosis-induced seizure | 0 | Yes | 1.8/2 | 30 |

Abbreviations: ATA, absolute atmospheres; DFU, diabetic foot ulcer; HBOT, hyperbaric oxygen therapy; ISSHL, idiopathic sensorineural hearing loss; NA, not available; OM, osteomyelitis; ORN, osteoradionecrosis.

## Hyperbaric oxygen therapy characteristics

Most patients underwent HBOT at a pressure of 2.0 ATA (18; 82%). Three patients (14%) received treatment at 1.8 ATA, and one patient at 2.4 ATA (5%). Overall, 634 HBOT sessions were completed by those included in this study, and each patient underwent an average of $29 \pm 17$ sessions.

## Incidence of seizures and adverse events

Among the 22 patients, only one patient (#14, Table 2) experienced a tonic-clonic seizure at the end of the 10th HBOT session (100 minutes since starting the treatment, after the second air break), which was conducted at 2.0 ATA with two air breaks delivered every 30 minutes. The patient experienced a seizure that lasted several minutes, resulting in loss of consciousness, tongue injury, and bleeding from the mouth. Oxygen supply to the chamber was replaced with air immediately after seizure onset, and treatment was aborted. The patient remained

unconscious after removal from the hyperbaric chamber and was transferred to the emergency department, where she recovered without further complications and was discharged home several hours later. The patient had similar seizures 6–12 months prior to HBOT but had been seizure-free since starting levetiracetam and lorazepam six months prior to starting HBOT. However, she disclosed that she was non-compliant with her anti-seizure medications and was not taking them as prescribed. She was assessed by a neurologist, and her EEG two months prior to HBOT showed no signs of seizure activity. Overall, the incidence of seizures was one in 634 treatments.

## Discussion

This study examined the safety profile of HBOT in a cohort of patients with a history of prior seizures. Over the course of 634 HBOT sessions among 22 patients, only one seizure was recorded, yielding an incidence of one in 634 treatments.

In the literature, there is considerable variability in the incidence of seizure during HBOT, ranging from 0.25 in 10,000 to 6 per 10,000 [4–13]. The variability of the rate could be explained by the clinical profile of patients included in the analysis and the differences in the treatment protocols.

Although a history of seizures is recognized by some authors as a risk factor for oxygen-induced seizures, the safety of HBOT in these patients has not been extensively evaluated (Table 3). Some studies did not specify whether patients with seizure histories were eligible for HBOT or included in the analyses [4–6, 8, 11]. Costa et al. excluded such patients from their analyses [13]. Yilidz et al. included data from centres that decline patients with seizures for HBOT [7]. Hadanny et al. treated only patients with a seizure-free interval of six months and a normalized EEG [10]. Sherlock et al. and Plafki et al., did not provide clinical data on patients with seizure histories [5, 12]. Heyboer et al. and Hadanny et al. offered limited information, reporting that at 2.0 ATA, the seizure risk may be negligible even in those with seizure histories [9, 10].

**Table 3. Incidence of HBOT related seizures, treatment pressures, and inclusion of patients with the history of seizure for data analysis.**

| Study (first author, year) | Total number of HBOT sessions | Seizure rate (%) | Treatment pressure (ATA) | Patients with seizure history |
|---|---|---|---|---|
| **Welslau, 1996** [4] | 107,264 | 0.015 | 2.4–2.9 | Not reported. |
| **Plafki, 2000** [5] | 11,376 | 0.035 | 2.4–2.5 | No history of epilepsy in 4 patients with oxygen-induced seizures. |
| **Hampson, 2003** [6] | 10,238 | 0.029 | 2–2.8 | Not reported. |
| **Yildiz, 2004** [7] | 80,679 | 0.002 | 2–2.8 | Not accepted for HBOT. |
| **Banham, 2011** [8] | 41,273 | 0.061 | 1.9–4 | Not reported. |
| **Heyboer, 2014** [9] | 23,328 | 0.047 | 2–2.8 | No statistically significant increased risk of oxygen-induced seizures. |
| **Hadanny, 2016** [10] | 62,614 | 0.011 | 1.5–2.8 | Only patients with seizure-free interval for six months and normalized EEG were accepted for HBOT. |
| | | | | No increased risk of oxygen-induced seizures. |
| **Jokinen-Gordon, 2017** [11] | 1,529,859 | 0.017 | 2–2.5 | Not reported. |
| **Sherlock, 2018** [12] | 96,670 | 0.039 | 2.4 | From 26 patients with HBOT-related seizures: one had epilepsy; four had a history of oxygen-induced seizures. No more details. |
| **Costa, 2019** [13] | 188,335 | 0.023 | 2.5–2.8 | Excluded from the analysis. |

Abbreviations: ATA, absolute atmospheres; HBOT, hyperbaric oxygen therapy.

To address the gaps in the existing literature and provide new insights into the safety of HBOT in patients with a history of seizures, our study included patients with a shorter seizure-free interval of three months, and one patient with ongoing focal aware seizures. This is in contrast with Hadanny et al. who only treated patients with a seizure-free period of six months. The Heyboer's trial lacked detailed safety data [9, 10].

Most patients in our cohort tolerated HBOT without complications, with only one patient experiencing a seizure during treatment (#14, Table 2). Differential diagnosis in this case includes underlying epilepsy that was worsening by poor compliance with antiseizure medications or oxygen induced seizure. While the exact cause of her seizure cannot be definitively determined, seizures were rare in this cohort of patients with a history of prior seizures.

The safety and low incidence of seizures during HBOT can be ascribed to several factors. Firstly, patient selection for HBOT in those with a history of seizures requires a comprehensive assessment. Stabilization and appropriate management of seizure disorders and comorbidities are important to minimizing risks during the therapy. Hadanny et al. found that the incidence of seizures is low when patients have a normalised EEG and a six months seizure-free period prior to HBOT [10]. In our study, patients with poorly controlled seizure disorders were not approved for HBOT, however, those with a 3-month seizure-free period were considered safe to initiate the therapy. One patient (#22, Table 2) with poorly controlled ongoing focal aware seizures due to radiation-induced brain necrosis was approved for HBOT following multidisciplinary discussion with the radiation oncologist and neurologist. The patient's condition improved after 30 uneventful HBOT sessions at 1.8 ATA. This case highlights the importance of understanding each patient's underlying seizure disorder and tailoring the HBOT protocol accordingly.

Treatment pressure and duration should be tailored to each patient. For patients with a history of seizures, lower pressures, such as 1.8 to 2.0 ATA, are generally recommended as they are associated with lower incidences of seizures compared to higher pressures ($\geq$2.4 ATA) [9, 10]. In this study, hyperbaric pressures were mostly at 2.0 ATA or lower (21; 95%).

Air breaks during treatment were another critical factor, as they help lower oxygen partial pressure and reduce the risk of oxygen toxicity [13]. Costa et al. previously demonstrated across over 180,000 HBOT sessions that introducing even a single 5-minute air break during a session could significantly reduce seizure frequency [13]. In this study, all patients received at least one 5-minute air break per session, with the majority receiving two to three air breaks (18; 82%). The frequency and duration of these breaks were adjusted based on each patient's needs. Additionally, the duration and number of HBOT sessions should be periodically reassessed, particularly in patients with chronic conditions, to minimize the risk of cumulative oxygen exposure increasing seizure susceptibility.

The method of oxygen delivery during HBOT is also important. It has been suggested that using a hood, as opposed to a mask, to deliver oxygen in multiplace chambers may increase the risk of seizures due to the higher partial pressure of oxygen and accumulation of carbon dioxide [7, 20]. The incidence of oxygen toxicity with hoods have previously been reported as 3 in 10,000 [7]. Inadequate removal of carbon dioxide can lead to hypercapnia, raising the risk of oxygen toxicity seizures [13, 16, 29]. All patients in the current study were treated in monoplace chambers pressurized with oxygen, minimizing the risk of carbon dioxide accumulation and potentially contributing to the low incidence of seizures.

In addition to these factors, individual physiological differences, such as age, comorbidities, and medications, may influence the oxygen toxicity seizures incidence [13]. In our patients' cohort, a total of 68% of the patients treated with HBOT were on antiseizure medications. Although, antiseizure medications do not entirely eliminate the risk of seizures, they have been

shown to reduce seizure onset, even under high treatment pressures [30]. This underscores the importance of optimizing antiseizure therapy before initiating HBOT.

Beyond seizure control, conditions that may lower the seizure threshold, such as hypoglycemia, electrolyte imbalances, and alcohol dependence, should be managed to further reduce the oxygen toxicity risk during treatment [4, 16, 31]. For instance, the incidence of seizures in general population has been shown to increase with age [32, 33]. This may be due to reduced resilience to oxidative stress and age-associated accumulation of reactive oxygen and nitrogen species [34]. Comorbid conditions are also influential in determining seizure risk. For example, patients with diabetes mellitus may experience hypoglycemic events during HBOT, which is a documented risk factor for seizures [35–37].

Patient safety can be further enhanced by educating hyperbaric staff for possible early signs of oxygen toxicity, such as vision changes, auditory changes, nausea, tingling, twitching, dizziness or vertigo, irritability, pallor, which can precede seizure onset [10]. Additionally, certain medications, such as antidepressants, tramadol, disulfiram, and cephalosporins, can lower the seizure threshold during HBOT [16]. Selective serotonin reuptake inhibitors, for example, can elevate serotonin levels in the brain, heightening neuronal excitability and reducing the seizure threshold [38–40]. Similarly, tramadol inhibits the reuptake of serotonin and norepinephrine, lowering the seizure threshold, especially in patients with a history of seizures [41–43]. These considerations emphasize the importance of individualized treatment plans that account for medication profiles and potential adverse interactions during HBOT.

## Limitations and strengths

The current study has several limitations, most of which are inherent to the study design. As the data were collected retrospectively, there is an increased risk of selection bias and potential for relevant risk factors to be missed. The study was also limited to two urban centers, reducing its generalizability, and patients were treated by different healthcare professionals, which could lead to inconsistencies in measurement and reporting. Additionally, all patients were treated in a monoplace chamber, thus the safety of HBOT in a multiplace chamber still requires further investigation.

Another significant limitation is the exclusion of 21 patients who were referred for HBOT but did not receive treatment. Among these, 11 patients refused to proceed with HBOT, while others were excluded due to factors such as poorly controlled seizure disorders, palliative state, severe confinement anxiety, multiorgan dysfunction, inability to commute, or referral to another center. The exclusion of these patients restricts the study's ability to fully assess the safety of HBOT in a broader population, particularly those at higher risk for adverse outcomes. Additionally, the small sample size further limits the power of quantitative analysis, given the low prevalence of patients with a history of seizures undergoing HBOT.

The strengths of the study comprise the inclusion of elective patients with various clinical indications for HBOT and the comprehensive range of clinical data reported, including medications, seizure histories, and details of the HBOT protocols.

## Conclusion

Although patients with a history of seizures can experience oxygen toxicity-induced seizures despite ongoing antiseizure therapy, its incidence remains very low. Given the relatively low incidence of seizures among patients with a history of seizures receiving HBOT, it may be considered in cases where the indication is strong, with close monitoring and assessment of seizure-related relative contraindications.

## Author Contributions

**Conceptualization:** Rita Katznelson.

**Data curation:** Subin Park, Anton Marinov, Elise Greer, Rita Katznelson.

**Formal analysis:** Subin Park, Rita Katznelson.

**Investigation:** Rita Katznelson.

**Methodology:** Rita Katznelson.

**Supervision:** Rita Katznelson.

**Writing – original draft:** Subin Park.

**Writing – review & editing:** Subin Park, Anton Marinov, Hance Clarke, Simone Schiavo, Elise Greer, George Djaiani, Jordan Tarshis, Rita Katznelson.

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
