## [Decision Letter · Decision Letter 0]

14 Nov 2024

PONE-D-24-43548Safety of hyperbaric oxygen therapy in non-emergent patients with the history of seizures: a retrospective cohort studyPLOS ONE

Dear Dr. Katznelson,

Thank you for submitting your manuscript to PLOS ONE. After careful consideration, we feel that it has merit but does not fully meet PLOS ONE’s publication criteria as it currently stands. Therefore, we invite you to submit a revised version of the manuscript that addresses the points raised during the review process.

**ACADEMIC EDITOR: ** **- **please do make minor correction as advised by our expert reviewer. I will made final decision as soon as I received your revised manuscript. 

We look forward to receiving your revised manuscript.

Kind regards,

Prof. Dr. Dragan Hrncic, MD, PhD 

Academic Editor

PLOS ONE

Journal Requirements:

Rita Katznelson, Anton Marinov and Hance Clarke are shareholders of the Rouge Valley Hyperbaric Medical Center. 

Reviewers' comments:

Reviewer's Responses to Questions

**Comments to the Author**

1. Is the manuscript technically sound, and do the data support the conclusions?

Reviewer #1: Yes

Reviewer #2: Yes

Reviewer #3: Partly

2. Has the statistical analysis been performed appropriately and rigorously? 

Reviewer #1: Yes

Reviewer #2: Yes

Reviewer #3: Yes

3. Have the authors made all data underlying the findings in their manuscript fully available?

Reviewer #1: Yes

Reviewer #2: Yes

Reviewer #3: Yes

4. Is the manuscript presented in an intelligible fashion and written in standard English?

Reviewer #1: Yes

Reviewer #2: Yes

Reviewer #3: Yes

5. Review Comments to the Author

Reviewer #1: Dear authors,

I've read with interest your manuscripts, which provides insight into the very difficult field of HBOT for epileptic patients. This subset of individuals is usually excluded from HBOT due to the belief that they will inevitably manifest seizures while under oxygen.

Introduction: very well balanced and comprehensive; just at line 53 please correct pre-exiting -> pre-exiSting

Methods: sound and well described

Results well described

Discussion and conclusion: balanced, with a thorough analysis and critical appraisal of the available literature, supporting the findings or defending and explaining differences.

Tables: appropriate, and provide the reader with all the needed data.

Reviewer #2: Thank you for the opportunity to review this submission, “Safety of hyperbaric oxygen therapy in non-emergent patients with the history of seizures: a retrospective cohort study”, for PLOS ONE. The article describes a retrospective cohort study of patients with pre-existing seizure disorders undergoing HBOT for a variety of indications.

The submission has several strengths. Among them, it addresses a topic of significant clinical uncertainty as seizure disorders are widely treated as a relative contraindication to HBOT (in the absence of compelling evidence). It builds on a thorough review of prior work, presented in table 3, and adds a—small but important—sample of patients to this body of literature. Finally, the report is generally well-written and its logic is easy to follow.

I believe that the article is essentially publishable, and offer several comments in the hopes that they may strengthen a final version:

1. This work is important, and the introduction could make a stronger case for why this is. Namely, that the referral of patients with seizure histories is not uncommon in hyperbaric units (especially given comorbid overlap of seizure disorders and some indicating conditions), and that denying these patients HBOT on the basis of no compelling evidence is a potential harm in and of itself.

a. The introduction could also emphasize, as the average reader of a general medical journal may not be immediately familiar with HBOT, that patients without a seizure history can also develop seizures as a result of oxygen toxicity.

2. The methods section could more clearly describe the inclusion criteria of patients for this retrospective review. In particular, it is stated that patients with “a documented history of seizures” were included, but could be specified whether this means a recurrent seizure disorder or—as Table 2 suggests—even a single lifetime, provoked seizure. The discussion section also presents some information relative to patient inclusion which does not fully align with the methods section: lines 195 and 209 state that patients were included with shorter seizure-free periods than in prior literature, and that patients with poorly-controlled seizures were excluded. Both are important pieces of information, but it should be stated whether these were predefined inclusion criteria for the retrospective review, or if these were general treatment principles of the participating units that shaped the population available to be reviewed.

3. In the results section, a short case vignette is provided to describe the single patient in this cohort who did experience a seizure during treatment. The circumstances of her seizure would benefit from more detail, if possible. In particular, the information presented in the paragraph beginning on line 200 belongs in this section, as it is very relevant to her seizure and the case. On line 170, it is unclear whether an unremarkable EEG resulted in her discontinuing her two anti-seizure medications or if this is just presented as evidence that her seizures were well-controlled. Finally, if possible, it would be useful to note when in her 10th treatment the seizure occurred (e.g., relative to either air break).

a. Also relevant to this patient’s case: it is challenging to suggest that the seizure was due to underlying epilepsy rather than oxygen toxicity, especially if it happened in the hyperbaric chamber and responded to replacement of oxygen with air. It may be more balanced to suggest that the cause of her seizure cannot be definitively proven but, in any case, seizures were rare in this cohort of patients with a history of prior seizures.

4. The presentation of the overall finding as a rate of 16 in 10,000 is awkward, given that there were fewer than 10,000 treatments described in the review. It is helpful to have a number comparable to previously-reported incidences (0.55 – 6 in 10,000 as presented in line 178), but given that this dataset describes a single seizure in a relatively small sample, I would suggest describing seizure rates as 1 in 22 patients and 1 in 634 treatments.

5. Finally, some minor comments:

a. The title should read “with a history” rather than “with the history”.

b. Lines 47/48: “despite” is not the right word, as the benefits and risks simply coexist (as with any treatment). Line 47 should read: “[…] barotrauma, claustrophobia and, in rare cases, seizures”.

c. Lines 52/53 should read: “[…] side effects of seizure-triggering [more aptly seizure threshold-lowering] medications, and pre-existing brain pathology”.

d. Line 75: there is a double period ending this sentence.

e. Line 109: for consistency with other subsections, an empty line should precede this paragraph.

f. Line 169: Levetiracetam and Lorazepam should not be capitalized.

g. Table 3: in the final column, several data fields end without punctuation or with a comma instead of a period.

Thank you again for the opportunity to review this submission, which I believe will be a valuable contribution to the clinical literature.

Reviewer #3: The study addresses an important question regarding the safety of hyperbaric oxygen therapy for patients with a history of seizures. The low incidence of seizures among the study population is promising; however, the absence of standardized HBOT protocols and the reliance on individualized treatment regimens may limit the generalizability of the study's conclusions.

The authors noted this limitation in their discussion; however, the conclusions would benefit from a more cautious tone. For example, in line 284, instead of, “Patients with known seizure disorders and seizure histories can safely complete HBOT courses for a variety of indications following predetermined protocols and careful monitoring,” I would suggest a more nuanced statement:

“Given the relatively low incidence of seizures among patients with a history of seizures receiving HBOT, it may be considered in cases where the indication is strong, with close monitoring and assessment of seizure-related relative contraindications.”

Additional minor comments:

Abstract: line 30: “...with five-minute air brakes”

I think this part of the sentence should be linked more closely to the following sentence.

Line 32: “One patient experienced a seizure event, and the overall incidence of seizures was 16 in 10,000 treatments. “

It may be clearer to state that the calculated rate of seizures was 10 per 10,000 sessions, rather than implying there were 16 individual cases of seizures.

Line 44: ref 2 is probably not the best reference for “HBOT is commonly indicated for a variety of conditions, including diabetic foot ulcers, chronic non-healing wounds, radiation injuries, and necrotizing infections, among others”

Line 80: can a study of 22 patients “optimize treatment protocols and improve patient outcomes”?

Line 202: “indicating that her seizure was likely due to underlying epilepsy rather than oxygen induced toxicity”

That might be an overly strong conclusion. It's noted that she was free of seizure when she met her neurologist two months prior to beginning HBOT.

Line 255: “certain medications.....can can lower the seizure threshold during HBOT”

There is a typo.

6. PLOS authors have the option to publish the peer review history of their article (what does this mean?). If published, this will include your full peer review and any attached files.

Reviewer #1: No

Reviewer #2: No

Reviewer #3: No

---

## [Author Response · Author response to Decision Letter 0]

4 Dec 2024

Responses to Comments

Thank you for your valuable feedback. We have worked to address each comment as described below. We believe the paper is now stronger after having made these changes. All changes have been highlighted in the manuscript. 

Changes to reference list

Ref#2 (Mathieu D, Marroni A, Kot J. Tenth European Consensus Conference on Hyperbaric Medicine: recommendations for accepted and non-accepted clinical indications and practice of hyperbaric oxygen treatment. Diving Hyperb Med 2017;47:24–32. https://doi.org/10.28920/dhm47.1.24-32.) was added to replace the previous reference to address Reviewer #3’s feedback.

Ref #27 (Yun C, Xuefeng W. Association between seizures and diabetes mellitus: a comprehensive review of literature. Curr Diabetes Rev 2013;9:350–4) was added as part of the revision that was made with respect to the comment made by Reviewer #2 on the introduction. 

Reviewer #1

Comment Response to comment Location in manuscript

Introduction: very well balanced and comprehensive; just at line 53 please correct pre-exiting -> pre-exiSting The error has been corrected. Page 4

Reviewer #2

Comment Response to comment Location in manuscript

This work is important, and the introduction could make a stronger case for why this is. Namely, that the referral of patients with seizure histories is not uncommon in hyperbaric units (especially given comorbid overlap of seizure disorders and some indicating conditions), and that denying these patients HBOT on the basis of no compelling evidence is a potential harm in and of itself. Thank you for this important feedback. We have expanded on the introduction to include the following:

“Referrals of patients with a history of seizures to hyperbaric units are not uncommon, especially due to the comorbidity between seizure disorders and conditions typically managed with HBOT, such as diabetic foot ulcers. Excluding these individuals from treatment without robust evidence of heighted risk may inadvertently lead to suboptimal care. Thus, understanding HBOT safety profile in patients with seizure history is essential for guiding clinical decision-making and obtaining informed patient consent for HBOT.” Page 5, line 89

The introduction could also emphasize, as the average reader of a general medical journal may not be immediately familiar with HBOT, that patients without a seizure history can also develop seizures as a result of oxygen toxicity. The following has been added to reflect the comment:

“Importantly, even patients without a prior history of seizures can develop oxygen-induced seizures.” Page 4, line 59

The methods section could more clearly describe the inclusion criteria of patients for this retrospective review. In particular, it is stated that patients with “a documented history of seizures” were included, but could be specified whether this means a recurrent seizure disorder or—as Table 2 suggests—even a single lifetime, provoked seizure. The discussion section also presents some information relative to patient inclusion which does not fully align with the methods section: lines 195 and 209 state that patients were included with shorter seizure-free periods than in prior literature, and that patients with poorly-controlled seizures were excluded. Both are important pieces of information, but it should be stated whether these were predefined inclusion criteria for the retrospective review, or if these were general treatment principles of the participating units that shaped the population available to be reviewed. The methods section has been revised to address the comments and described in further detail. The follow changes have been added:

“A history of seizures was defined as one or more provoked or non-provoked seizure events obtained from medical records. Additionally, the participating hyperbaric centres approved HBOT for patients who had been seizure-free for at least three months, and those with poorly controlled seizure disorders were determined on a case-by-case basis but generally not approved to initiate therapy.”

 Page 6, line 117

In the results section, a short case vignette is provided to describe the single patient in this cohort who did experience a seizure during treatment. The circumstances of her seizure would benefit from more detail, if possible. In particular, the information presented in the paragraph beginning on line 200 belongs in this section, as it is very relevant to her seizure and the case. We have added the information previously in line 200 to this section:

“Among the 22 patients, only one patient (#14, Table 2) experienced a tonic-clonic seizure at the end of the 10th HBOT session (100 minutes since starting the treatment, after the second air break), which was conducted at 2.0 ATA with two air breaks delivered every 30 minutes” Page 11, line 197

On line 170, it is unclear whether an unremarkable EEG resulted in her discontinuing her two anti-seizure medications or if this is just presented as evidence that her seizures were well-controlled. We clarified this as:

“[…]she disclosed that she was non-compliant with her anti-seizure medications and was not taking them as prescribed.” Page 11, line 206

Finally, if possible, it would be useful to note when in her 10th treatment the seizure occurred (e.g., relative to either air break). Please see our reply above. 

Also relevant to this patient’s case: it is challenging to suggest that the seizure was due to underlying epilepsy rather than oxygen toxicity, especially if it happened in the hyperbaric chamber and responded to replacement of oxygen with air. It may be more balanced to suggest that the cause of her seizure cannot be definitively proven but, in any case, seizures were rare in this cohort of patients with a history of prior seizures. This is certainly an important point to clarify. We have made the following changes to reflect this comment:

“Differential diagnosis in this case includes underlying epilepsy that was worsening by poor compliance with antiseizure medications or oxygen induced seizure. While the exact cause of her seizure cannot be definitively determined, seizures were rare in this cohort of patients with a history of prior seizures.” Page 14, line 251

The presentation of the overall finding as a rate of 16 in 10,000 is awkward, given that there were fewer than 10,000 treatments described in the review. It is helpful to have a number comparable to previously-reported incidences (0.55 – 6 in 10,000 as presented in line 178), but given that this dataset describes a single seizure in a relatively small sample, I would suggest describing seizure rates as 1 in 22 patients and 1 in 634 treatments. Thank you for this important feedback. We have changed the reporting of the incidence of seizures to one in 22 patients or one in 634 treatments where applicable. Page 2, 10, 11

The title should read “with a history” rather than “with the history”. The following change has been made to the title:

Safety of hyperbaric oxygen therapy in non-emergent patients with a history of seizures: a retrospective cohort study Page 1

Lines 47/48: “despite” is not the right word, as the benefits and risks simply coexist (as with any treatment). 

 The following change has been made:

“However, the therapy is not without potential side effects, which can include ear barotrauma, claustrophobia, and in rare cases, seizures.” Page 4, line 54

Line 47 should read: “[…] barotrauma, claustrophobia and, in rare cases, seizures”. The following change has been made:

“However, the therapy is not without potential side effects, which can include ear barotrauma, claustrophobia, and, in rare cases, seizures.” Page 4, line 54

Lines 52/53 should read: “[…] side effects of seizure-triggering [more aptly seizure threshold-lowering] medications, and pre-existing brain pathology”. The following change has been made:

“[…] side effects of medications that lower the seizure threshold, […].” Page 4, line 61

Line 75: there is a double period ending this sentence. The extra period has been removed. Page 5

Line 109: for consistency with other subsections, an empty line should precede this paragraph. Line has been added preceding this paragraph. Page 7

Line 169: Levetiracetam and Lorazepam should not be capitalized. Capitalizations have been removed. Page 11, line 206

Table 3: in the final column, several data fields end without punctuation or with a comma instead of a period. Punctuations have been added where missing. Page 12

Reviewer #3

Comment Response to comment Location in manuscript

The authors noted this limitation in their discussion; however, the conclusions would benefit from a more cautious tone. For example, in line 284, instead of, “Patients with known seizure disorders and seizure histories can safely complete HBOT courses for a variety of indications following predetermined protocols and careful monitoring,” I would suggest a more nuanced statement:

“Given the relatively low incidence of seizures among patients with a history of seizures receiving HBOT, it may be considered in cases where the indication is strong, with close monitoring and assessment of seizure-related relative contraindications.” Thank you for this valuable feedback. We certainly agree with the concern raised. We have replaced the specified sentence with the suggestions. Page 17, line 348

Abstract: line 30: “...with five-minute air brakes”

I think this part of the sentence should be linked more closely to the following sentence.

Line 32: “One patient experienced a seizure event, and the overall incidence of seizures was 16 in 10,000 treatments. “ Thank you for this feedback. We have made the following changes to reflect this comment:

“In total, 634 HBOT sessions were administered to 22 patients in monoplace chambers with five-minute air breaks, and one patient experienced a seizure event.” Page 2, line 34

It may be clearer to state that the calculated rate of seizures was 10 per 10,000 sessions, rather than implying there were 16 individual cases of seizures. Thank you for this feedback. After receiving similar feedback from another reviewer, we have made changes to the reporting of the incidence of seizures to one in 634 treatments where applicable. Page 2, 10, 11

Line 44: ref 2 is probably not the best reference for “HBOT is commonly indicated for a variety of conditions, including diabetic foot ulcers, chronic non-healing wounds, radiation injuries, and necrotizing infections, among others” The reference has been replaced. Page 4, line 53

Line 80: can a study of 22 patients “optimize treatment protocols and improve patient outcomes”? Thank you for your feedback. We have made the following changes in respect to this point:

“The study seeks to provide insights into its safety, which may inform future research and contribute to refining treatment protocols.” Page 6, line 102

Line 202: “indicating that her seizure was likely due to underlying epilepsy rather than oxygen induced toxicity”

That might be an overly strong conclusion. It's noted that she was free of seizure when she met her neurologist two months prior to beginning HBOT. This is certainly an important point to clarify. We have made the following changes to reflect this comment:

“Differential diagnosis in this case includes underlying epilepsy that was worsening by poor compliance with antiseizure medications or oxygen induced seizure. While the exact cause of her seizure cannot be definitively determined, seizures were rare in this cohort of patients with a history of prior seizures.” Page 14, line 251

Line 255: “certain medications.....can can lower the seizure threshold during HBOT” 

There is a typo. The following change has been made:

“[…] can lower [...].” Page 16, line 317

---

## [Decision Letter · Decision Letter 1]

2 Jan 2025

Safety of hyperbaric oxygen therapy in non-emergent patients with the history of seizures: a retrospective cohort study

PONE-D-24-43548R1

Dear Dr. Katznelson,

We’re pleased to inform you that your manuscript has been judged scientifically suitable for publication and will be formally accepted for publication once it meets all outstanding technical requirements.

Kind regards,

Prof. Dr. Dragan Hrncic, MD, MSc, PhD

Academic Editor

PLOS ONE

Additional Editor Comments (optional):

Reviewers' comments:

Reviewer's Responses to Questions

**Comments to the Author**

1. If the authors have adequately addressed your comments raised in a previous round of review and you feel that this manuscript is now acceptable for publication, you may indicate that here to bypass the “Comments to the Author” section, enter your conflict of interest statement in the “Confidential to Editor” section, and submit your "Accept" recommendation.

Reviewer #3: All comments have been addressed

2. Is the manuscript technically sound, and do the data support the conclusions?

Reviewer #3: Yes

3. Has the statistical analysis been performed appropriately and rigorously? 

Reviewer #3: Yes

4. Have the authors made all data underlying the findings in their manuscript fully available?

Reviewer #3: Yes

5. Is the manuscript presented in an intelligible fashion and written in standard English?

Reviewer #3: Yes

6. Review Comments to the Author

Reviewer #3: (No Response)

7. PLOS authors have the option to publish the peer review history of their article (what does this mean?). If published, this will include your full peer review and any attached files.

Reviewer #3: No

---

## [Editor Report · Acceptance letter]

3 Jan 2025

PONE-D-24-43548R1 

PLOS ONE

Dear Dr. Katznelson, 

I'm pleased to inform you that your manuscript has been deemed suitable for publication in PLOS ONE. Congratulations! Your manuscript is now being handed over to our production team.

Kind regards, 

on behalf of

Professor Dragan Hrncic 

Academic Editor

PLOS ONE